# Survey of Tetrodotoxin in New Zealand Bivalve Molluscan Shellfish over a 16-Month Period

**DOI:** 10.3390/toxins12080512

**Published:** 2020-08-10

**Authors:** Michael J. Boundy, Laura Biessy, Brian Roughan, Jeane Nicolas, D. Tim Harwood

**Affiliations:** 1Cawthron Institute, 98 Halifax Street East, Nelson 7010, New Zealand; michael.boundy@cawthron.org.nz (M.J.B.); laura.biessy@cawthron.org.nz (L.B.); 2New Zealand Food Safety Science & Research Centre, Massey University, Palmerston North 4442, New Zealand; 3Ministry for Primary Industries–Manatū Ahu Matua, PO Box 2526, Wellington 6140, New Zealand; roughanb@gmail.com (B.R.); jeane.nicolas@mpi.govt.nz (J.N.)

**Keywords:** HILIC-MS/MS, emerging marine toxin, saxitoxin, shellfish, tetrodotoxin

## Abstract

Tetrodotoxin (TTX) is a heat-stable neurotoxin typically associated with pufferfish intoxications. It has also been detected in shellfish from Japan, the United Kingdom, Greece, China, Italy, the Netherlands and New Zealand. A recent European Food Safety Authority (EFSA) scientific opinion concluded that a level of <0.044 mg TTX/kg in marine bivalves and gastropods, based on a 400 g portion size, does not result in adverse effects in humans. There have been no reports of human illness attributed to the consumption of New Zealand shellfish containing TTX. To obtain a greater understanding of its presence, a survey of non-commercial New Zealand shellfish was performed between December 2016 and March 2018. During this period, 766 samples were analysed from 8 different species. TTX levels were found to be low and similar to those observed in shellfish from other countries, except for pipi (*Paphies australis*), a clam species endemic to New Zealand. All pipi analysed as part of the survey were found to contain detectable levels of TTX, and pipi from a sampling site in Hokianga Harbour contained consistently elevated levels. In contrast, no TTX was observed in cockles from this same sampling site. No recreationally harvested shellfish species, including mussels, oysters, clams and tuatua, contained TTX levels above the recommended EFSA safe guidance level. The levels observed in shellfish were considerably lower than those reported in other marine organisms known to contain TTX and cause human intoxication (e.g., pufferfish). Despite significant effort, the source of TTX in shellfish, and indeed all animals, remains unresolved making it a difficult issue to understand and manage.

## 1. Introduction

Tetrodotoxin (TTX) is a potent neurotoxin that has been responsible for many human intoxications and deaths around the world, primarily from consumption of pufferfish (fugu). The distribution of TTX and its analogues in the environment is remarkably diverse, being found in a variety of organisms from both marine and terrestrial environments [1,2,3]. The source of TTX is still controversial and not definitively proven, although the dominant hypothesis is that it is of microbial origin [4,5]. There have been reports of TTX in 21 species of bivalves and edible gastropods from 10 countries since the 1980’s [6]. Increasing reports of the detection of TTX in aquaculture species such as bivalve molluscs has drawn considerable recent attention to the toxin, reinvigorating scientific interest and questioning whether regulation is required. TTX has similar sodium channel blocking action and potency to the paralytic shellfish toxin group (saxitoxin (STX) and analogues) but is structurally dissimilar. The maximum permissible level adopted in most countries for paralytic shellfish toxins in bivalve molluscs is 0.8 mg STX.2HCl eq/kg. In March 2017, the European Food Safety Authority (EFSA) published their scientific opinion on the risk to public health related to the presence of TTX and analogues in marine bivalves and gastropods. They concluded that a concentration below 0.044 mg TTX eq/kg, based on a large portion size of 400 g, was considered not to result in adverse effects in humans [7]. Liquid chromatography with tandem mass spectroscopy (LC–MS/MS) methods were determined to be the most suitable for the specific identification and quantification of TTX and its analogues. Japan used a maximum portion size of 1000 g, to determine that pufferfish are safe to consume below a TTX concentration of <10 MU/g (2 mg/kg) [8,9].

There have been reports of TTX in shellfish from Japan, New Zealand, the United Kingdom, Greece, China, Italy and the Netherlands. In contrast, several recent surveys looking for TTX in shellfish from the Iberian Peninsula did not find detectable levels. This included mussels (*Mytilus galloprovincialis*), oysters (*Crassostrea gigas*) and clams (*Ruditapes philippinarum* and *Donax* spp.) from Portugal [10], and various bivalve species from the Northwest and East of Spain [11,12]. In 1993, TTX was reported at concentrations of up to 40 MU/g (~8 mg TTX/kg) in the digestive glands of Japanese scallop (*Patinopecten yessoensis*). TTX was confirmed as being present using a variety of analytical techniques that included high-performance liquid chromatography with fluorescence detection and fast atom bombardment-mass spectrometry (FAB-MS) after the toxin was partially purified from scallop tissue [13]. In New Zealand, the first report was in pipi (*Paphies australis*), an endemic clam, collected from the North Island in 2011. Concentrations up to 0.8 mg/kg were reported in these samples with analyses being performed using two separate methods that employed LC-MS/MS [14]. One approach analysed intact toxin, while the other monitored a TTX-C9 base derivatisation product generated by TTX dehydration under highly alkaline conditions. More recent studies on TTX-contaminated pipi have identified organ level differences, with the siphon found to contain significantly higher TTX levels than other organs of the shellfish [15]. Depuration was observed in TTX-contaminated pipi maintained in captivity on a toxin-free diet and a concentration gradient has also been observed with higher levels observed in shellfish from the warmer northern latitudes of New Zealand [16]. In 2012 in Greece, during official shellfish monitoring for the presence of marine biotoxins, a series of unexplained positive mouse bioassay screens were observed. This was not attributed to any regulated toxin and analysis by LC-MS showed the presence of TTX at levels of up to 0.223 mg/kg in Mediterranean mussels (*Mytilus galloprovincialis*) [17]. Analysis of Greek library samples, mussels and *Venus verrucosa* (clams), between 2006–2012 showed TTX at levels between 0.061–0.194 mg/kg. In England, TTX was first reported in blue mussels (*Mytilus edulis*) and Pacific oysters (*Crassostrea gigas*) from southern England in 2014, with levels of up 0.12 mg/kg observed in the small sample set (*n* = 29) [18]. In this study, shellfish harvested from two sites on the south coast of England were screened for TTX using a hydrophilic interaction liquid chromatography (HILIC)-MS/MS method and its presence was confirmed using LC-MS/MS after derivatisation under alkaline conditions to the TTX-C9 base. Several TTX analogues were observed at low levels in the samples. A s more comprehensive study of various shellfish species collected from southern England between 2014 and 2016 reported a maximum level of 0.253 mg/kg a Pacific oyster sample [19]. In Italy (Sicily) in 2018, TTX was detected for the first time in 14 out of the 25 shellfish samples analysed. The species analysed were mussels (*Mytilus galloprovincialis*) and clams (*Venerupis decussata*) with a maximum level of only 0.0064 mg TTX eq/kg observed. TTX has also been reported in The Netherlands with a maximum level of 0.253 mg/kg in Pacific oysters (*Crassostrea gigas*) and 0.101 mg/kg in blue mussels (*Mytilus edulis*) from production areas [20]. TTX analogues were also monitored for using high-resolution LC-MS and were not observed, except for 4-*epi*-TTX in a single sample. In China, during 2015, as part of the validation for a new LC-MS method, Manila clams (*Ruditapes philippinarum*) purchased from markets in China were analysed for the presence of TTX and trace levels were observed [21]. A similar observation was made in another Chinese study looking at TTX in aquatic products, with low µg/kg levels observed in clams (*R. philippinarum*), blue mussels (*M. edulis*), hard-shell mussels (*Mytilus corucus*) and Chinese razor clam (*Sinonovacula constricta*) [22]. The maximum level observed was 0.016 mg/kg in a Chinese razor clam.

In the present study, a survey on TTX in non-commercial New Zealand bivalves was performed over a 16-month period to determine the prevalence of this toxin and help determine if it represents a food safety risk to shellfish consumers. Samples analysed were collected as part of the non-commercial shellfish marine biotoxin monitoring programme administered by the New Zealand Ministry for Primary Industries (MPI), New Zealand Food Safety.

## 2. Results

### 2.1. Survey Results 

In total, 766 shellfish samples were analysed over the 16-month period (Table 1). Sample matrices analysed comprised Greenshell^TM^ mussels (*Perna canaliculus*; 63%), tuatua (*Paphies subtriangulata*; 28%) and pipi (*Paphies australis*; 6%), with fewer than 10 samples in total of blue mussels (*Mytilus edulis*), clams (unspecified), cockles (*Austrovenus stutchburyi*), Pacific oysters (*Crassostrea gigas*) and rock oysters (*Saccostrea glomerata*). There was no TTX detected in the majority of samples (69%, Table 1). A further 27% of samples had detectable TTX but at levels below the recommended safe guidance level reported in the 2017 EFSA scientific opinion (0.002–0.044 mg/kg). Another 4% of samples had TTX levels greater than the safe guidance level (≥0.044 mg/kg), with all of these being pipi. In fact, all pipi tested as part of this survey contained detectable TTX levels. These percentages are likely to be biased by the unequal numbers of samplings for each species but provide a valuable insight into TTX levels in many recreationally harvested shellfish species.

Most samples were taken from sites located in the North Island as historical information shows these as the most at-risk areas for harmful algal blooms and hence why they are routinely monitored for regulated marine toxins. It was possible to overlay a map of New Zealand with the sampling location and number of shellfish samples tested for TTX (Figure 1). 

Most pipi samples included in the survey came from Koutu Point, which is in the Hokianga Habour (Figure 2). Consistently high TTX levels were detected in pipi from this site over the time monitored (Figure 3A; blue dots). In contrast, cockles collected from the same site at the same time contained no detectable TTX (Figure 3A; black dots). To assess TTX level variability between individuals, pipi were collected from Tauranga harbour in February 2017 and 12 individual shellfish were analysed separately for TTX in addition to the usual pooled homogenate from a minimum of 12 individuals. Similar TTX levels were observed between individual shellfish with a precision <20% relative standard deviation (RSD) (Figure 3B). A pooled pipi homogenate generated gave a TTX level of 0.15 mg/kg, which is in line with the median level from the analysis of individuals.

Additional shellfish samples were sourced from the Hokianga Harbour as part of a PhD project (collected on 27 October 2017). These were collected outside of the main survey and were also tested for the presence of TTX. Samples included juvenile mussels and oysters, snails, pipi and cockles from areas close to the Koutu Point sampling site and from the harbour entrance >5 km away. Low levels of TTX were observed in all the samples, ranging from 0.003–0.04 mg/kg. None of these additional samples exceeded the safe guidance level reported in the EFSA scientific opinion.

### 2.2. Analysis of Archive Shellfish Samples

To determine if the presence of TTX in New Zealand shellfish is a recent phenomenon, archived frozen samples (2001–2003 *n* = 18, 2007–2009 *n* = 9) were obtained from long-term storage and analysed. Of these samples, 8 contained detectable TTX levels. The highest TTX concentrations in the samples taken between 2001 and 2003 was 0.019 mg/kg, and between 2007 and 2009 was 0.021 mg/kg. No archive samples tested were pipi. The detection rate of TTX in the archived frozen samples was consistent with the detection rate observed in the survey samples collected between December 2016 and March 2018 (31%).

### 2.3. Presence of Tetrodotoxin (TTX) Analogues

Selected shellfish samples that contained TTX above an arbitrary threshold level of 0.02 mg TTX/kg were re-analysed for the presence of known TTX analogues using a targeted TTX acquisition method. Quantitation of the other analogues was off TTX and assumed an equivalent response. In all cases TTX was found to be the most abundant analogue, accounting for >98% of the total TTX analogues present. Other TTX analogues were observed but in most cases were present at too low a concentration to allow quantitation. As a representative example, see Figure 4 showing the presence of TTX (0.19 mg TTX/kg) in a pipi sample and trace detections of structurally related analogues. The assignment of the TTX analogues was made based on comparison of retention time against published findings [23] and a naturally contaminated, and well characterized, flatworm quality control sample. 

### 2.4. Outlier Sample from Outside Survey Period

Whilst establishing the methodology to monitor TTX in shellfish, several non-commercial samples being assayed for paralytic shellfish toxins were also monitored for TTX. A Greenshell^TM^ mussel sample from Browns Bay in the Hauraki Gulf was found to contain TTX at a level of 1.6 mg/kg. This site had been sampled to determine the geographical spread of an *Alexandrium pacificum* paralytic shellfish toxin bloom event in the Mahurangi Harbour (Northland, Figure 2). Paralytic shellfish toxins were also present in this sample at 0.4 mg STX·2HCl eq/kg (the CODEX standard for reporting saxitoxin-group toxins is in saxitoxin dihydrochloride equivalents per kg), representing half the regulatory limit for this toxin class. To assess the contribution of TTX to the total toxicity of the sample, it was subjected to a mouse bioassay (AOAC959.08) that targets these toxins and was found to be above regulatory limit at 1.3 mg STX·2HCl eq/kg. The result from the mouse bioassay is consistent with the expected underestimation of TTX toxicity due to its slower death response time than saxitoxin [24]. Shellfish sampled from the same site pre- and post-this result contained only trace TTX levels, demonstrating the rapid appearance and disappearance of TTX in shellfish from this site (Table 2).

## 3. Discussion

Low levels of TTX (>0.002 mg/kg) were detected in 31% of all non-commercial New Zealand shellfish tested during the survey period, with most samples being below the reporting limit of the method. No seasonal trends were observed at any sites sampled multiple times over the survey period. TTX analogues were observed in shellfish found to contain TTX, but only at much lower levels relative to TTX itself. No recreationally harvested shellfish that were tested during the survey, including mussels, oysters, clams and tuatua, contained TTX levels above the recommended EFSA safe guidance level of 0.044 mg/kg. The observation of TTX in New Zealand shellfish does not represent a new phenomenon as archive samples contained toxin at similar levels and frequency to shellfish tested during the survey. 

TTX levels observed in individual pipi were similar to that of the pooled sample, indicating that TTX levels were similar in all pipi from that site. At times, pipi had levels that exceeded the recommended EFSA guidance level, but not the Japanese limit, including all samples taken from the Hokianga Harbour site. Cockles collected from the same site at the same time did not contain TTX. This represents a particularly interesting finding. Recent research on TTX-contaminated pipi identified tetrodotoxin in all the dissected organs but that the siphons contained significantly higher levels [16]. It has been hypothesized that *P. australis* contains unique TTX-binding proteins in the siphon similar to those found in the puffer fish (*Fugu pardalis*) [25] or the crab (*Hemigrapsus sanguineus*) [26], that allowed selective retention of the toxin within different organs. Hokianga Harbour is estuarine in nature and one of the northern most sampling sites, which means that it experiences warmer temperatures than the other sampling sites further south. Recent research has identified a latitudinal gradient for TTX levels in pipi, supporting the warmer water hypothesis [16]. The association of higher TTX levels in shellfish from warmer environments raises concerns that this toxin may become an increasing human health concern as the global climate warms.

TTX has been reported in many terrestrial and marine species, including bivalve shellfish, although its origin remains unclear. Accumulation from the diet, whether from bacteria or another source, is an attractive hypothesis supported by the observation that cultured pufferfish are found to be non-toxic and that pipi know to contain TTX depurate the toxin when fed a toxin-free diet. However, TTX levels found in bacteria and marine sediments are low, and production by bacterial species has still not been demonstrated. Many scientists believe bacteria are the source of TTX in shellfish, and this sentiment is reflected in recent EFSA opinion on the presence of TTX in bivalve shellfish, which begins: “TTX and its analogues are produced by marine bacteria and have been detected in marine bivalves and gastropods from European waters” [7]. Bacterial cultures have been reported to contain low TTX concentrations and are suggested to be the ultimate biosynthetic origin of the toxin. However, these results remain controversial and are disputed due to poor specificity of the methods of analysis used, or extremely low levels observed when more specific methods of analysis are employed. More conclusive data is needed to unequivocally determine the exogenous or endogenous source of TTX in shellfish.

When assessing sample toxicity, co-occurrence of TTX with paralytic shellfish toxins needs to be considered. This is because both TTX and STX bind to voltage-gated sodium channels and our research has shown that they have additive toxicological effects [24]. The PSP mouse bioassay, which is still used for regulatory monitoring in some countries, is not able to distinguish TTX from saxitoxin group toxins (paralytic shellfish toxins). This is important as the presence of paralytic shellfish toxins in shellfish is regulated whereas the presence of TTX currently is not. Shellfish containing paralytic shellfish toxins below the regulatory action limit of 0.8 mg STX·2HCl eq/kg could be found to be above the regulatory threshold for this toxin class when using the PSP mouse bioassay, if TTX is also present. This situation, although likely a rare occurrence, has been observed in a non-commercial shellfish sample analysed prior to the main survey. A single mussel sample was found to contain 1.6 mg of TTX/kg and paralytic shellfish toxins at 0.4 mg STX·2HCl eq/kg (half the regulatory limit). When subjected to the PSP mouse bioassay, the toxicity of the sample was determined to be 7278 MU/kg or 1.3 mg STX·2HCl eq/kg, which is above the paralytic shellfish toxin regulatory limit of 0.8 mg STX.2HCl eq/kg. This demonstrates that even though TTX levels in New Zealand shellfish are typically low, there is potential for sporadic high levels. The factors that resulted in the high TTX level observation in this single sample are unresolved, although it is known that marine biota exists in NZ that contain elevated TTX and could be potential vectors (e.g., grey side-gilled sea slug [27] and flatworms [28]). In New Zealand, chemical analytical methods are used for all routine regulatory control of marine biotoxins in shellfish. For paralytic shellfish toxins, a LC-MS/MS method is used that allows specific identification of the various paralytic shellfish toxin analogues samples and simultaneous monitoring of TTX. 

Having an accurate assessment of TTX toxicity and an understanding of the mechanism of TTX accumulation in marine foodstuffs is important for managing the potential risk to consumers. Existing data on TTX toxicity by oral administration are limited, with a wide range of results reported in the literature. Most of the information available relates to acute toxicity through intraperitoneal injection to mice, and this route of administration is of questionable relevance, given that seafood is consumed orally rather than injected. To address this we have recently generated robust toxicity information for TTX via oral administration (gavage and voluntary feeding) and demonstrated that the toxicities of STX and TTX are additive [24]. Data gaps still exist. For example, toxicity information for other TTX analogues (such as 11-oxo-TTX) is needed, as they are potentially equipotent with TTX and have been documented to be dominant analogues in some species of crab [29]. In New Zealand, the grey side-gilled sea slug (*Pleurobranchaea maculata*) is the most well-known TTX-containing organism and has been reported to lay highly toxic eggs [30]. The presence of these organisms, or their eggs, in shellfish harvesting areas could make them a possible vector of the toxin. In addition, many marine worm species also contain high TTX concentrations and they could potentially contaminate bivalve shellfish. This mechanism of toxin transfer represents a plausible explanation for the elevated TTX levels observed in the outlier mussel sample from Browns Bay. The likelihood of this possibility is heightened by the fact that the Browns Bay site is close to where toxic *P. maculata* have been found in the past. 

## 4. Conclusions

TTX was detected in about one-third of samples tested during the survey of New Zealand shellfish. The levels were low and below the recommended EFSA safe limit, except for pipi, a common surf clam. All pipi tested during the survey contained TTX and a site was identified in the Hokianga Harbour where consistently elevated levels were observed. Recent research on pipi has identified that the siphon contains TTX levels significantly higher than other parts of the shellfish. Despite these findings it remains important to determine the source of TTX in bivalve shellfish, and indeed other marine and terrestrial organisms. 

## 5. Materials and Methods

### 5.1. Shellfish Sampling

Shellfish samples received weekly (December 2016–March 2018; Cawthron Institute, Nelson) for paralytic shellfish toxin testing as part of the MPI administered non-commercial marine toxin shellfish monitoring programme were also analysed for TTX. In total there were 56 sampling sites from around New Zealand (Table 3) with a total of 766 samples analysed for TTX during the survey. 

During the survey, pipi sourced from the Hokianga Harbour (Northland, New Zealand) were found to contain TTX levels well above shellfish from other areas. With this site not routinely being monitored for marine toxins, a request was made to increase the frequency of sampling for the duration of this study. From March 2017, fortnightly sampling of the pipi bed (Koutu Point, Hokianga Harbour) was performed. Individual pipis (*n* = 12) from one sampling event in February 2017 from the Tauranga Harbour (Bay of Plenty) were tested, in addition to a pooled sample, to determine TTX variability between individuals.

In addition, to assess whether the presence of TTX in New Zealand shellfish is a recent phenomenon a subset of 27 archived (2001–2003 *n* = 18, 2007–2009 *n* = 9) shellfish homogenate samples were retrieved from frozen storage and analysed for the presence of TTX. Samples that are in frozen storage are typically from routine monitoring activities and contain detectable levels of regulated marine toxins. Very few pipi samples were in the archive.

### 5.2. Sample Preparation and Analysis of TTX by HILIC-MS/MS

A HILIC-MS/MS method was used that was developed as a collaboration with the Cawthron Institute and the Centre for Environment, Fisheries and Aquaculture Science (Cefas; UK) scientists for routine regulatory monitoring of the paralytic shellfish toxin group [31,32], which could also be expanded to monitor TTX [33]. The limit of reporting for the method was 0.002 mg TTX/kg. Certified tetrodotoxin (TTX) material (6.75 ± 0.24 µg/g) was purchased from the National Research Council Canada (NRC, Halifax, NS, Canada). When TTX was observed in shellfish samples, it was possible to re-analyse the sample extract using a targeted TTX acquisition method to allow monitoring of a range of TTX analogues. As no reference material is currently available for the various known TTX analogues, it was not possible to accurately quantify them. Therefore, the concentration of each analogue was semi-quantified using an assumed relative response factor of 1. This will introduce a source of error, but in the absence of reference material it is the only option currently available to allow semi-quantification of TTX analogues.

Briefly, 5.0 ± 0.1 g of homogenised shellfish tissue was weighed into a centrifuge tube followed by the addition of 5 mL of 1% acetic acid. The mixture was vortex-mixed before being placed in a boiling water bath for 5 min. Samples were then cooled for 5 min in an ice slurry, before further vortex mixing. Samples were centrifuged at 3,200× *g* for 10 min before pipetting a 1 mL aliquot into a 5 mL polypropylene tube and adding 5 μL of 25% ammonia. For sample cleanup, Supelclean ENVI-Carb 250 mg/3 mL solid phase extraction (SPE) cartridges (Sigma-Aldrich, St. Louis, MO, USA) were conditioned at 6 mL/min using 3 mL of 20% acetonitrile +0.25% acetic acid, before the addition of 3 mL of 0.025% ammonia. A 400 μL aliquot of the acetic acid extract was loaded onto the cartridge, followed by washing with 700 μL of deionized water. Sample extracts were eluted with the addition of 2 mL of 20% acetonitrile +0.25% acetic acid and collected. SPE eluents were vortex-mixed prior to dilution of 100 μL aliquots with 300 μL acetonitrile.

### 5.3. Animals 

Female Swiss albino mice (18–22 g) were bred at AgResearch, Ruakura, New Zealand. The mice were housed individually during the experiments and were allowed unrestricted access to food (Rat and Mouse Cubes, Speciality Feeds Ltd., Glen Forrest, Australia) and water. All experiments were approved by the Ruakura Animal Ethics Committee established under the Animal Protection (code of ethical conduct) Regulations Act., 1987 (New Zealand), Project Number 14005, approval date 6 October 2016.

## Figures and Tables

**Figure 1 toxins-12-00512-f001:**
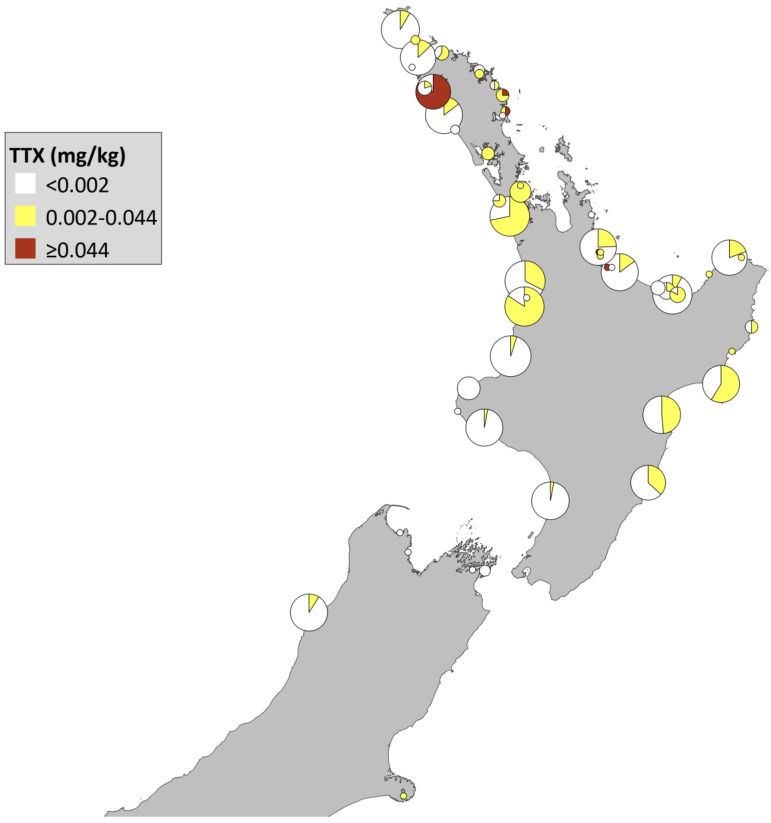
Location of sampling sites and tetrodotoxin levels observed in shellfish collected from around New Zealand between December 2016 and March 2018. The size of the circle corresponds to number of samples from a site (max = 41). Colouration within each circle shows the proportion of samples that fall within each of the three defined TTX levels. No samples were taken for the part of New Zealand not shown in the map.

**Figure 2 toxins-12-00512-f002:**
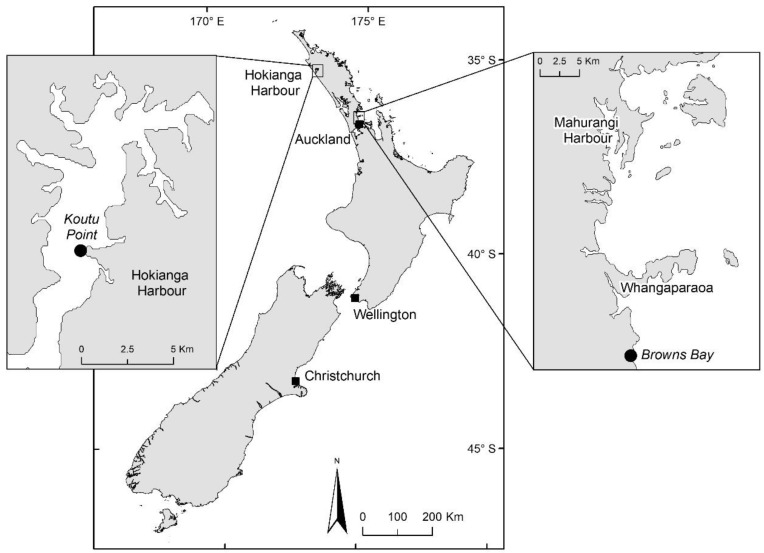
Location of Koutu Point and Browns Bay sampling sites in the north island of New Zealand.

**Figure 3 toxins-12-00512-f003:**
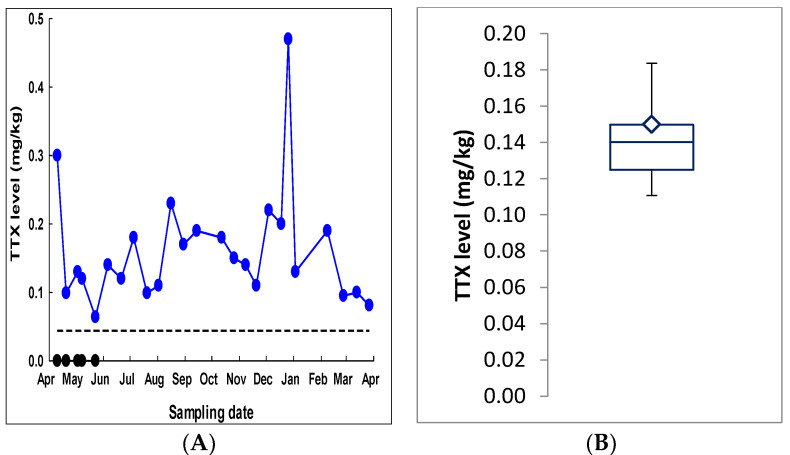
(**A**) Tetrodotoxin levels in pipi and cockles sourced from Koutu Point in the Hokianga Harbour between April 2017 and April 2018. Blue dots = pipi; black dots = cockles. Dotted line represents European Food Safety Authority (EFSA) safe guidance level (0.044 mg/kg). (**B**) Box and whisker plot showing tetrodotoxin levels measured in 12 individual pipi sourced from Tauranga harbour in February 2017. Shown is the median, interquartile range, 5th and 95th percentiles. The diamond represents the result from the pooled homogenate.

**Figure 4 toxins-12-00512-f004:**
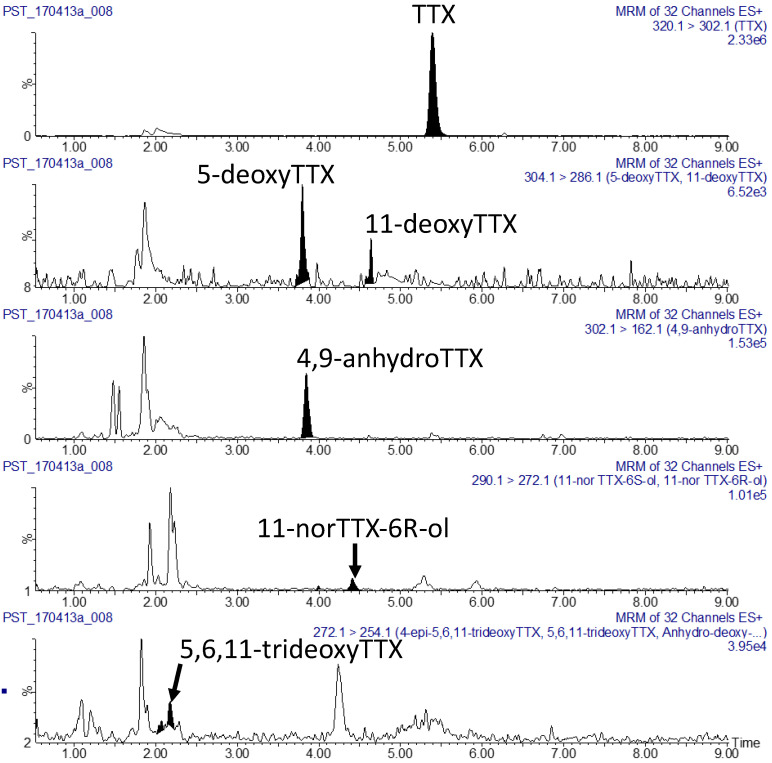
Observed TTX analogues from a Koutu point pipi sample (0.19 mg TTX/kg).

**Table 1 toxins-12-00512-t001:** Summary of non-commercial shellfish samples analysed for tetrodotoxin (TTX) (December 2016–March 2018).

Shellfish Species	TTX Level (Mg/Kg)			
	<0.002	0.002–0.044	≥0.044	Total
Blue mussel	4	3	0	7
Clams	1	0	0	1
Cockle	7	1	0	8
Greenshell^TM^ mussel	318	162	0	480
Pacific oyster	0	1	0	1
Pipi	0	15	29	44
Rock oyster	3	6	0	9
Tuatua	194	22	0	216
Total	527 (69%)	210 (27%)	29 (4%)	766

**Table 2 toxins-12-00512-t002:** TTX levels in samples taken from Browns Bay and nearby sites from May–July 2016.

Date Sampled.	Shellfish	Distance from Browns Bay (Km)	TTX Level (Mg/Kg)
11 May 2016	Pacific oyster	11	0.002
1 June 2016	Greenshell^TM^ mussel	11	0.02
6 June 2016	Greenshell^TM^ mussel	11	0.01
6 June 2016	Greenshell^TM^ mussel	5	<0.002
19 June 2016	Greenshell^TM^ mussel	0	0.004
3 July 2016	Greenshell^TM^ mussel	0	1.6
11 July 2016	Pacific oyster	11	0.004
17 July 2016	Rock oyster	11	0.003

**Table 3 toxins-12-00512-t003:** Sampling site details and summary of TTX levels observed in non-commercial shellfish samples tested during the survey period.

Site		TTX Level (Mg/Kg)					
	Site Code	Average	Maximum	<0.002	0.002–0.044	> = 0.044	Total Sample #
Mangonui Harbour	SA006	0.007	0.025	2	3	0	5
The Bluff-90 Mile Beach	SA025	0.000	0.007	33	3	0	36
Waipapakauri	SA027	0.001	0.006	27	4	0	31
Tapeka Point	SA030	0.015	0.023	0	2	0	2
Taurua-Reef Point	SA036	0.000	0.000	1	0	0	1
Black Rocks (Bay of Islands)	SA040	0.000	0.000	3	0	0	3
Houhora Wharf	SA129	0.013	0.023	0	2	0	2
Oakura	SB001	0.001	0.002	1	1	0	2
Parua Bay	SB007	0.000	0.000	1	0	0	1
Pataua	SB008	0.038	0.045	0	1	1	2
Whananaki	SB032	0.028	0.045	0	3	1	4
Browns Bay	SC032D	0.009	0.033	0	11	0	11
Whangaparaoa Peninsula	SC032F	0.010	0.010	0	1	0	1
Tairua Harbour	SD012	0.000	0.000	1	0	0	1
Waihi Beach	SD017	0.001	0.010	25	8	0	33
Tauranga Harbour-Upper	SD018	0.000	0.000	1	0	0	1
Tauranga Harbour-Lower	SD021	0.048	0.048	0	0	1	1
Papamoa Beach	SD025	0.001	0.008	29	5	0	34
Pukehina Beach	SD028	0.000	0.000	36	0	0	36
Bowentown	SD030	0.013	0.013	0	1	0	1
Katikati-Tauranga Harbour	SD031	0.041	0.041	0	1	0	1
Katikati-Tauranga Harbour	SD031P	0.150	0.150	0	0	1	1
Kauri Point	SD031S	0.038	0.038	0	1	0	1
Whakatane Heads	SD032	0.001	0.002	5	2	0	7
Waiotahi	SD036	0.007	0.012	1	5	0	6
Ohope Beach	SD037	0.000	0.004	35	3	0	38
Whangaparaoa	SD041	0.001	0.007	25	6	0	31
Thornton	SD042	0.000	0.000	5	0	0	5
Te Kaha	SD050	0.002	0.002	0	1	0	1
Tolaga Bay Wharf	SE001	0.005	0.017	2	2	0	4
Mahia, Opoutama	SE006	0.003	0.018	14	20	0	34
Pania Reef	SE007	0.002	0.015	18	17	0	35
Taikorai Rocks-Porangahau	SE010A	0.002	0.013	19	11	0	30
Lottin Point	SE019	0.003	0.003	0	1	0	1
Gisborne Wharf	SE028	0.004	0.004	0	1	0	1
Muriwai, West Coast	SF009	0.006	0.015	1	3	0	4
Cornwallis (Manukau Hbr)	SF015	0.004	0.022	11	28	0	39
Raglan	SF016	0.001	0.007	27	13	0	40
Kawhia	SF017	0.006	0.024	6	32	0	38
Mohakatino	SF018	0.000	0.003	39	2	0	41
Oakura Beach	SF020	0.000	0.000	13	0	0	13
Koutu Point (Hokianga Hbr)	SF021	0.134	0.470	5	0	25	30
Tinopai (Kaipara Hbr)	SF026	0.006	0.010	0	4	0	4
Maunganui Bluff	SF029	0.001	0.007	29	5	0	34
Mitimiti	SF031	0.000	0.002	4	1	0	5
Aotea Harbour	SF033	0.000	0.000	1	0	0	1
Bayleys Beach	SF156	0.000	0.000	2	0	0	2
Tapu Bay–Tasman Bay	SG006	0.000	0.000	1	0	0	1
Wedge Point	SG023	0.000	0.000	1	0	0	1
Onapua Bay	SG123	0.000	0.000	3	0	0	3
Pohara	SG313	0.000	0.000	1	0	0	1
Ohawe Beach	SH001	0.000	0.002	33	1	0	34
Foxton	SH002	0.000	0.003	34	1	0	35
Lower Kina Road	SH023	0.000	0.000	1	0	0	1
The Kaik	SI004	0.002	0.002	0	1	0	1
Cape Foulwind	SJ004	0.000	0.006	31	3	0	34

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
