# Peer review of "Survey of Tetrodotoxin in New Zealand Bivalve Molluscan Shellfish over a 16-Month Period"

_toxins, 2020, doi:10.3390/toxins12080512_

Round 1

Reviewer 1 Report

This paper mainly deals with the results of TTX survey in bivalves collected from various parts of New Zealand during Dec. 2016 – Mar. 2018. Although several reports on TTX levels in an endemic species pipi have already been published, this is the first comprehensive study using a large number of shellfish samples from various species. In addition, a comparison with archive samples, and periodic survey of TTX levels in pipi at a fixed point and comparison with another species are also conducted, which will provide valuable information on the food safety risk of TTX. I recommend the authors to make minor revisions based on the following comments/questions.

  1. Keywords: It may be better to add “saxitoxin”
  2. 4-epi-TTX (L75): → 4-epiTTX
  3. Sample matrices ----- being pipi (L89-96): Table 1 shows the number of individuals, but it is explained in percentage in the text. As this is a little confusing, why not writing both the number of individuals and the percentage in either Table 1or the text?
  4. Flatworm quality control sample (L151): It is better to cite a relevant reference(s).
  5. Table 4: I don’t think a hyphen is needed between “deoxy” or “anhydro” and “TTX”. Please check the exact description of each analogue.
  6. Alexndrium pacificum (L158): In italics.
  7. PSP (L158): What does PSP represent?
  8. saxitoxin (L164): → STX
  9. Table 2: Please indicate in the footnote what the hyphen in the “Distance from Browns Bay” column represents.
  10. Low levels of TTX were detected (>0.002 mg/kg) (L169): “Low levels of TTX (>0.002 mg/kg) were detected” may be better.
  11. tetrodotoxin (L183): → TTX
  12. Table 3: Regarding the “Site” column, it seems better to use lowercase except for the first letter. How about mentioning in the footnote that the average was calculated by considering <0.002 as 0. As Table 3 contains the results, it may be better to move to Results section or cite as Appendix. Please consider.
  13. In addition ----- the archive (L266-270): It would be better to mention the sample of PhD project and outlier sample from outside survey period at this part.
  14. Please describe the ethical approval for mouse bioassay.
  15. References: There are several parts where the scientific name is not italicized, the use of upper/lower case letters is not correct, or the journal name is not abbreviated. Please check.

Reviewer 2 Report

The paper represents and interesting contribution to the actual impact of TTXs in New Zealand. It is, in general, well written, reasonably well documented, understandable and the main conclusions are supported by the obtained data.

Some references are clearly missing, for example, dealing with oral toxicity of TTX https://doi.org/10.3390/toxins9030075 , doi:10.3390/toxins12050312, or with its distribution among species and areas doi:10.3390/toxins11060331, doi:10.3390/jmse7070232, doi: 10.3390/md16030081. doi:10.3390/md17010028  can also be taken into account. Some affirmations are not supported by references as lines 194-195, lines 200-201 and line 203.

The subject of lines 218-222 seems to be unrelated to that of the paragraph in which it is included. In can be moved to another location or the paragraph split.

The bias introduced in the study by the large differences in sample number is superficially commented in the text. A more detailed explanation of the consequences should be included, and, additionally, it has to be taken into account when drawing conclusions about spatial or temporal variability (in several locations in the discussion section).

The procedure to obtain TTX-C9 base is lacking from the material and methods section.
